# Resonant perovskite solar cells with extended band edge

Jiangang Feng [1,2,6], Xi Wang [1,2,6], Jia Li[1,2], Haoming Liang [1,2], Wen Wen[3], Ezra Alvianto[1,2], Cheng-Wei Qiu [4], Rui Su [3,5] & Yi Hou [1,2] ✉

Tuning the composition of perovskites to approach the ideal bandgap raises the single-junction Shockley-Queisser efficiency limit of solar cells. The rapid development of narrow-bandgap formamidinium lead triiodide-based perovskites has brought perovskite single-junction solar cell efficiencies up to 26.1%. However, such compositional engineering route has reached the limit of the Goldschmidt tolerance factor. Here, we experimentally demonstrate a resonant perovskite solar cell that produces giant light absorption at the perovskite band edge with tiny absorption coefficients. We design multiple guide-mode resonances by momentum matching of waveguided modes and free-space light via Brillouin-zone folding, thus achieving an 18-nm band edge extension and 1.5 mA/cm² improvement of the current. The external quantum efficiency spectrum reaches a plateau of above 93% across the spectral range of ~500 to 800 nm. This resonant nanophotonics strategy translates to a maximum EQE-integrated current of 26.0 mA/cm² which is comparable to that of the champion single-crystal perovskite solar cell with a thickness of ~20 μm. Our findings break the ray-optics limit and open a new door to improve the efficiency of single-junction perovskite solar cells further when compositional engineering or other carrier managements are close to their limits.

Compositional engineering to narrow the bandgap of perovskite towards ideal bandgap of 1.34 eV raises the upper efficiency limit of perovskite solar cells[1–3]. So far, the majority of reported champion perovskite single-junction solar cells are based on formamidinium lead triiodide (FAPbI₃)[4–9]. FAPbI₃ possesses an optical bandgap of 1.48 eV which is both the narrowest and the closest to the ideal bandgap among lead-based perovskites[5–8]. However, such compositional engineering route has reached the limit of the Goldschmidt tolerance factor, especially for the selection of A-site cation[10]. For the selection of X-site, iodide already presents the narrowest bandgap among halide anions[10,11]. On the other hand, the incorporation of other cations such as tin or germanium on the B-site often incurs inferior stability[12].

To continue improving perovskite solar cell efficiencies, it is essential to further extend the band edge of perovskite to approach the ideal bandgap of single-junction solar cell. Single-crystal perovskite solar cells (~20 μm in thickness) present narrower bandgaps than that of the submicron-thick (~0.70 μm) polycrystalline thin film counterpart[13–15]. The narrowing of the photovoltaic (PV) bandgap from 1.54 to 1.49 eV stems from the increased absorption length of near-band-edge photons in single-crystal perovskites[14]. Although it suggests that increasing crystal thickness will extend the band edge further, the fabrication of large-area thick single crystal perovskite is challenging, and it requires higher quality perovskites to minimize charge recombination through 20-μm-thick single crystals. To mitigate the low absorption of submicron perovskite films at the band edge, a strong

[1]Department of Chemical and Biomolecular Engineering, National University of Singapore, Singapore 117585, Singapore. [2]Solar Energy Research Institute of Singapore (SERIS), National University of Singapore, Singapore 117574, Singapore. [3]Division of Physics and Applied Physics, School of Physical and Mathematical Sciences, Nanyang Technological University, Singapore 637371, Singapore. [4]Department of Electrical and Computer Engineering, National University of Singapore, Singapore 117583, Singapore. [5]School of Electrical and Electronic Engineering, Nanyang Technological University, Singapore 639798, Singapore. [6]These authors contributed equally: Jiangang Feng, Xi Wang. ✉e-mail: yi.hou@nus.edu.sg

light confinement needs to be introduced at this spectral regime. Unfortunately, this is impossible for the traditional light-management routes using textures and diffraction which operate in ray-optics regime[16,17].

Sub-wavelength resonant nanostructures, such as gratings and photonic crystals, sustain strong light confinement and light-matter interaction. These resonant structures have demonstrated promising applications in nano-lasers, nonlinear optics, and photodetectors[18–20]. For light management in solar cells, resonant structures provide a wave-optics solution for surpassing ray-optics limit[16,17,21–23]. Optical resonances have been successfully incorporated in high-index silicon and III-V materials for efficient ultrathin solar cells[24–26], but are still challenging for perovskite solar cells.

We report the integration of optical resonances with perovskite solar cells and extension of the perovskite band edge to maximize the short circuit current ($J_{sc}$) without sacrificing other PV parameters. To overcome the low absorption of FAPbI$_3$-based perovskite at the band edge, we demonstrate that multiple guided-mode resonances (GMRs), arising from Brillouin-zone (BZ) folding in a supercell grating, sustain giant light confinement and absorption enhancement near perovskite band edge. In this resonant solar cell comprising a supercell grating and a waveguide slab, normally incident free-space light can be coupled into the perovskite waveguide. This resonant solar cell narrows the PV bandgap by up to 35 meV, which corresponds to 18 nm redshift of band edge, and thus leading to an

EQE-integrated $J_{sc}$ up to 26.0 mA/cm$^2$. The strong band-edge light absorption, together with preserved efficient carrier transport in submicron perovskites, produces a power conversion efficiency of 24.4%. In comparison to ray-optics approaches employing texture and diffraction, the resonant perovskite solar cell manifests pronounced improvements in photocurrent and efficiency, attributed to its unprecedented giant band-edge light absorption (Supplementary Table 1). This folded resonance strategy can be implemented on low-index, low-absorption-coefficient semiconductors, paving the way for the development of efficient ultrathin single-junction and tandem solar cells.

## Results and discussion
### Optical resonance for narrowing PV bandgap
The balance between photocarrier thermalization and sunlight absorption identifies an optimum bandgap of 1.34 eV in Shockley-Queisser (S-Q) limit. For single-junction solar cells, GaAs with a bandgap of 1.42 eV produces a record efficiency of 29.1%[4]. Likewise, among lead-halide perovskites, FAPbI$_3$ with the narrowest bandgap of 1.48 eV outperforms its perovskite counterparts, including MAPbI$_3$ and CsPbI$_3$ (Fig. 1a, Supplementary Table 2 for $J$-$V$ parameters). Although optical bandgap of FAPbI$_3$ is the lower bound of lead-based perovskites, PV bandgap derived from external quantum efficiency (EQE) spectrum of the devices is 40–80 meV larger than the optical bandgap in state-of-the-art FAPbI$_3$ solar cells[5–9,27,28]. Beyond optical bandgap, PV bandgap

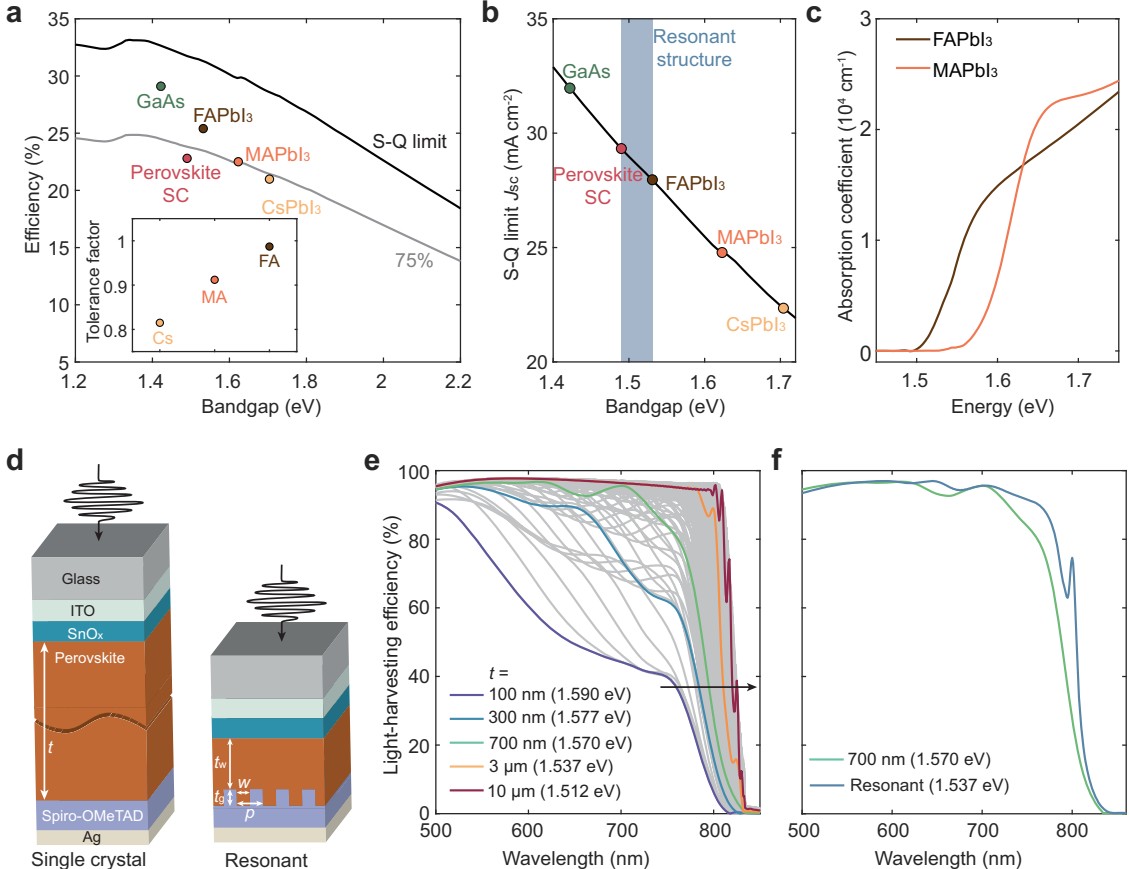

**Fig. 1 | PV bandgap narrowing for efficient perovskite solar cells. a** Theoretical Shockley-Queisser (S-Q) limit efficiency (black line) and record cell efficiency of GaAs[4], perovskite single crystals (perovskite SC)[14], and thin films (FAPbI$_3$[8], MAPbI$_3$[41], and CsPbI$_3$[42]) as a function of PV bandgap. Gray line is 75% of the limit. Inset illustrates tolerance factors of Pb perovskites with different A cations. **b** S-Q limit short circuit current as a function of PV bandgap. **c** Absorption coefficients of FAPbI$_3$ and MAPbI$_3$ perovskites near band edge. **d** Schematic of single-crystal and

resonant perovskite solar cells. For resonant solar cells, a perovskite waveguide slab with a thickness $t_w$ = 680 nm and a grating with a period $p$ = 535 nm and a height $t_g$ = 250 nm and a ridge width $w$ = 214 nm are coupled to support multiple guide-mode resonances. **e** Simulated LHE spectra of mixed-cation perovskite (optical bandgap of 1.556 eV) solar cells with perovskite thickness ranging from 100 nm to 10 μm. **f** Simulated LHE spectra of perovskite solar cells consisting of 700 nm thin film and resonant structures illustrate resonance modes extending band edge.

represents the figure of merit for evaluating SQ limit as it considers the deviation of near-band-edge absorption in practical semiconductors[2,29,30]. In contrast to silicon and GaAs counterparts which have nearly no difference between the optical and PV bandgaps, this 40−80 meV energy mismatch in perovskites provides a large room to further boost $J_{sc}$ in perovskite single-junction solar cells by optical resonances, especially in FAPbI$_3$-based perovskites (Fig. 1b). FAPbI$_3$-based perovskite is thermally stable and close to the ideal bandgap, rendering FAPbI$_3$ the most attractive perovskite layer for single-junction perovskite solar cells. However, the absorption coefficient of FAPbI$_3$ perovskites is relatively lower compared to MAPbI$_3$, which makes it even more crucial to improve the optical absorption near the band edge of FAPbI$_3$ perovskites (Fig. 1c).

To realize the band edge extension, we first simulate the light-harvesting efficiency (LHE) in single-crystal and resonant solar cells using finite-difference time-domain (FDTD) method (see Method for details). In our simulations, LHE is defined as the fraction of photons absorbed by the perovskite layer, while excluding photons that are reflected or absorbed by other layers such as electrodes, transport layers, and glass. It is important to note that LHE is conceptually equivalent to EQE, under the assumption of ideal conversion of absorbed photons to charge carriers, as well as ideal charge injection and collection[31].

A full "iodide" perovskite FA$_{0.95}$Cs$_{0.05}$PbI$_3$ with an optical bandgap of 1.556 eV is employed for this simulation (see refractive index, extinction coefficient and absorption spectrum in Supplementary Fig. 1). The LHE spectra are directly evaluated based on n-i-p devices with single-crystal or optically resonant perovskite light-absorption layers, while SnO$_x$ and spiro-OMeTAD are utilized as electron- and hole-transport layers respectively (Fig. 1d). By assuming unity quantum efficiency from absorbed photons to generated carriers, we can extract PV bandgap from first derivative of simulated LHE spectra, i.e. $dLHE/d\lambda$ (Supplementary Fig. 2), where $\lambda$ is the light wavelength. LHE spectra of single-crystal cells illustrate continuous redshift of PV bandgap from 1.590 to 1.510 eV with the increase of perovskite thickness $t$ from 100 nm to 10 μm, demonstrating that more efficient optical absorption near band edges can narrow PV bandgap (Fig. 1e). The PV bandgaps of thick perovskites are even narrower than the optical bandgap, which is caused by tiny but non-zero absorption coefficient below its optical bandgap (Supplementary Fig. 1d)[32,33].

To extend the perovskite band edge, multiple strong optical resonances should be introduced at wavelengths ranging from 790 to 830 nm. This is mainly because the double-pass optical absorption, evaluated by $2\alpha t$ where $\alpha$ is absorption coefficient, is below unity at wavelengths above 790 nm for a perovskite layer with a thickness of 700 nm (Supplementary Fig. 3). Moreover, the thickness of perovskite should be kept below 1 μm to guarantee efficient charge extraction. Considering these two principles, we designed a perovskite slab with a thickness $t_w$ of 680 nm and a grating with a height $t_g$ of 250 nm, a period $p$ of 535 nm, and a ridge width $w$ of 214 nm. The slab and the grating are coupled to introduce multiple GMRs at the band edge. This resonant nanostructure supports GMRs near perovskite band edge, thus leading to a high LHE of 75% at the wavelength of 800 nm and narrowing of PV bandgap to 1.537 eV (Fig. 1f).

## GMRs in perovskite resonant structures

For strong light in-coupling at perovskite band edge, we rationally design perovskite resonant structures. Firstly, a single-cell grating, which has one ridge in each unit cell, is coupled with a waveguide slab to host GMRs (Fig. 2a, top). The waveguide slab sustains guided modes with waves propagating via total internal reflection, but these modes located at high momentum cannot be excited by free-space light radiation. A diffraction grating provides an in-plane momentum which is determined by the grating period. This allows the coupling of free-space light into waveguide slab via GMRs. Such mechanism enables

sufficient light absorption even under relatively low extinction coefficient due to the strong light confinement. Secondly, a supercell grating with five non-identical ridges in each unit cell is designed to introduce additional GMRs and increase photonic density of state near band edge, thanks to the folding of Brillouin zone (BZ) in these supercell structures (Fig. 2a, bottom).

These resonant nanostructures can be efficiently fabricated in a large area through a nanoimprinting process (see details in Method and scheme of fabrication process in Supplementary Fig. 4). Scanning electron microscopy (SEM) images illustrate conformal transfer of single-cell and supercell gratings onto perovskites (Fig. 2b). The nanoimprinting method enables large-scale fabrication of resonant structures for solar-cell applications. Grating structures ca. 1 cm$^2$ in area (Fig. 2c, inset) can be successfully transferred onto perovskite in three minutes and the template is reusable. These single-cell and supercell gratings are homogeneous in a long range, which is demonstrated by low-magnification SEM images with clear fast Fourier transform (FFT) patterns (Supplementary Fig. 5). X-ray diffraction results illustrate nearly no difference between the thin film and the perovskite resonant structures (Fig. 2c), which suggests that the nanoimprinting process does not introduce additional damage to the perovskites.

On the basis of high-quality perovskite resonant structures, we seek to rigorously demonstrate multiple GMRs for light confinement. Angle-resolved reflectance spectra together with eigenfrequency calculations depict the photonic bandstructure of these gratings. For a single-cell grating, we can observe three photonic bands under transverse magnetic (TM) polarization, among which only one mode is near zero angle and perovskite band edge (Fig. 2e). Calculated photonic bandstructure is in line with the measurement results and illustrates three modes with symmetric/anti-symmetric field profiles (Supplementary Fig. 6). In contrast to the single-cell counterpart, two photonic bands are introduced at BZ center near the band edge of a perovskite supercell grating (Fig. 2f). Bandstructure calculation of a supercell grating illustrates replication of photonic bands in momentum space and folding of photonic bands from high momenta to BZ center at zero angle (Fig. 2d, Supplementary Fig. 7). Therefore, increased photonic density of state is realized in a supercell grating through band folding, which provides more channels for in-coupling of free-space light into waveguides.

Confocal photoluminescence (PL) mapping further evidences enhancement of light-matter interactions after incorporating resonant structures (Fig. 2g, h). Compared to a perovskite thin film with weak and grain-to-grain inhomogeneous PL emissions, pronounced enhancement of PL can be observed in perovskite resonant structures (Fig. 2g, h, Supplementary Fig. 8). The material quality improvement or lattice strain[34] for stronger PL can be ruled out by time-resolved PL measurements, which shows similar PL lifetimes for the perovskite thin film and resonant structure (Fig. 2i). We also find that the steady-state PL shows not only enhanced PL intensity but also a redshift of PL spectrum in perovskite resonant structures (Supplementary Fig. 9). Angle-resolved PL spectra further demonstrate bright PL emissions from the photonic bands at wavelengths above 800 nm (Supplementary Fig. 10). Therefore, we rationalize that this PL enhancement is a result of guided-mode resonances near the band edge, which sustain strong near fields and efficient photon emission channels[35].

## Resonant solar cells for PV bandgap narrowing

With the fundamental understanding into GMRs for light confinement, we combine these resonant structures with solar cells. We simulate the light absorption of resonant solar cells based on a full device with nanostructured perovskite absorbers, electrical transport layers and electrodes. By calculating the dispersion of LHE spectra, overlap of transverse electric (TE) and TM resonance modes at 790−820 nm can be observed for a resonant perovskite solar cell with a single-cell

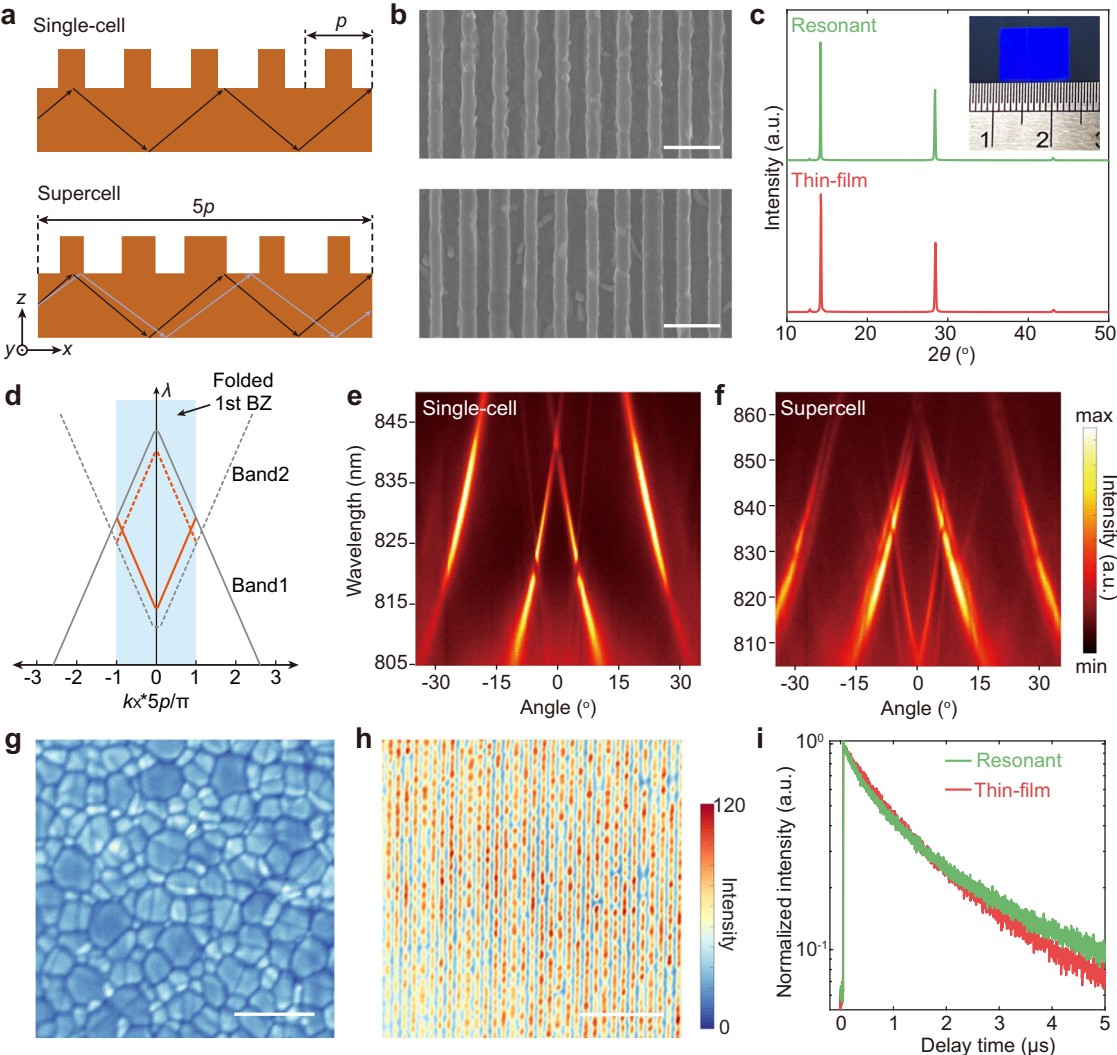

**Fig. 2 | GMRs in perovskite nanophotonic structures. a** Schematic illustration of GMRs in single-cell and supercell gratings. Supercell gratings with five non-identical ridges present folded Brillouin zone (BZ) and additional GMRs at perovskite band edge compared to a single GMR in single-cell counterparts. **b** SEM images of single-cell (top) and supercell (bottom) gratings. **c** XRD diagram of perovskite resonant nanostructure and thin film. **d** Schematic representation of band folding in supercell gratings. Gray lines illustrate two photonic bands in a single-cell grating.

In a supercell grating with one-fifth of BZ size, photonic bands at high momenta fold back to BZ center, yielding additional bands at the BZ center (displayed as orange lines). Angle-resolved reflectance spectra of single-cell (**e**) and supercell (**f**) gratings. Confocal PL mappings of a perovskite thin film (**g**), and resonant nanostructure (**h**). **i** Time-resolved PL of perovskite thin film and resonant nanostructure. "Resonant" nanostructure refers to the supercell grating. Scale bars, **b**, 1 μm, **g**, **h**, 5 μm. a.u., arbitrary units.

grating ($p$ = 510–540 nm) (Fig. 3a). GMR is demonstrated by the magnetic-field distribution under TM polarization (Supplementary Fig. 10). Owing to the device configuration with back-reflection, guided modes are mainly confined in the spiro-OMeTAD layer and coupled with GMRs in perovskites under orthogonal TE polarization. With an optimized grating period of 535 nm, PV bandgap is redshifted to 1.537 eV (807 nm). To further improve the band edge extension effect, we demonstrate a much higher photonic density of state near band edge using a supercell grating (Fig. 3b). In this structure, the dispersion of LHE as a function of the grating period and wavelength indicates additional GMRs which are caused by folding of photonic bands from high momenta to the center of the first BZ (Fig. 3b). We also observed two resonance modes under both TE and TM polarizations (Supplementary Fig. 11). Benefiting from the additional resonance modes, PV bandgap is theoretically narrowed to 1.525 eV (813 nm).

To further illustrate PV-bandgap narrowing, we calculated the photocarrier generation rate excited by sub-optical-bandgap light with photon energy ranging from 1.480 to 1.556 eV (Fig. 3c). For resonant solar cells based on single-cell and supercell gratings, strong

photocarrier generation is observed and its spatial distribution is consistent with the field profiles (Supplementary Figs. 10, 11), indicating that GMRs sustain strong absorption in sub-bandgap regime. In contrast, low photocarrier generation rate is observed in thin-film solar cells.

To evaluate resonance modes for solar-cell performances, we compare EQE spectra of solar cells based on thin film, single-cell and supercell gratings (Fig. 3d). The PV bandgap can be determined by the Gaussian distribution of d$EQE$/d$E$, where $E$ is the photon energy. The PV bandgap is narrowed from 1.570 eV (790 nm) of thin-film PV to 1.542 eV (804 nm) and 1.535 eV (808 nm) of resonant solar cells based on single-cell and supercell gratings respectively (Fig. 3e). This PV bandgap narrowing allows more photon absorption at the band edge regime. The best resonant solar cells based on single-cell and supercell gratings show EQE-integrated $J_{sc}$ of 25.6 and 26.0 mA/cm² respectively. In contrast, thin-film devices show the highest $J_{sc}$ of 24.5 mA/cm² only. The statistics of EQE-integrated $J_{sc}$ of the three samples are in line with this trend, in which average $J_{sc}$ improvement of 1.5 mA/cm² is achieved (Fig. 3f). To rule out the possible contribution of light scattering to

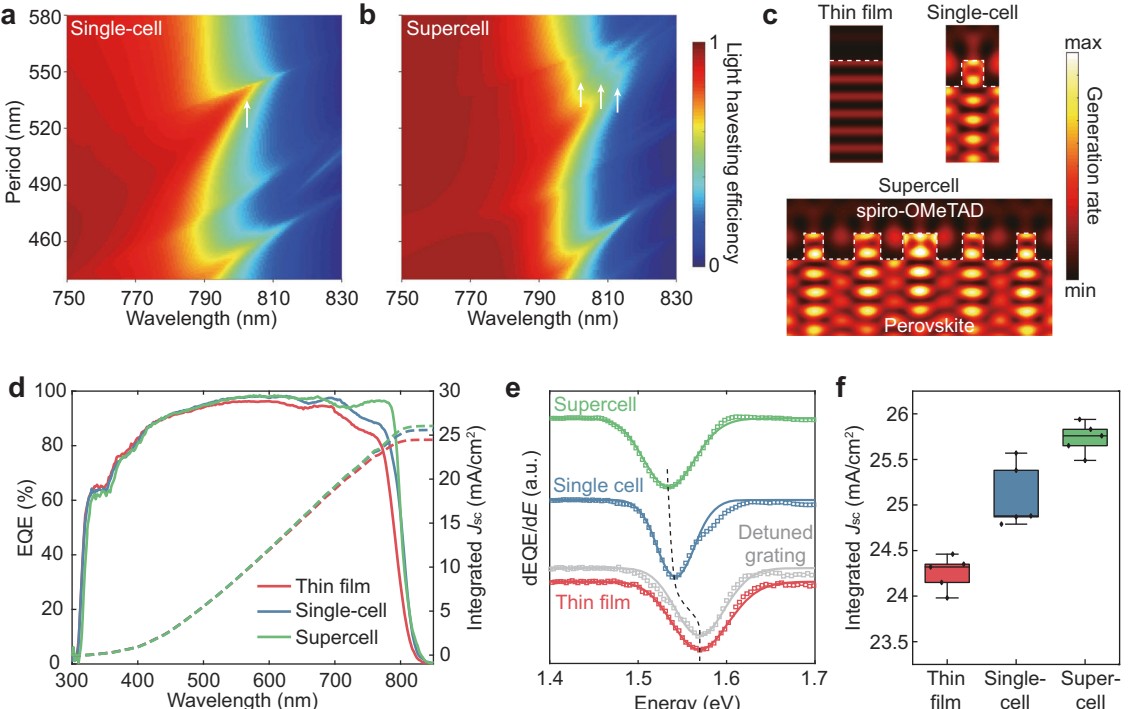

**Fig. 3 | Resonant solar cells with extended PV band edge.** Simulated dispersion of light-harvesting efficiency as a function of grating period in resonant solar cells with (**a**) single-cell, (**b**) supercell gratings. **c** Simulated maps of photocarrier generation rates in sub-bandgap (1.480-1.556 eV) spectral regime. EQE and integrated currents (**d**), dEQE/d*E* (**e**), and statistics of EQE integrated currents (**f**), of perovskite solar cells based on thin film, single-cell and supercell gratings. Black dashed line in (**e**) is a guide to the eye. a.u., arbitrary units.

PV-bandgap narrowing, we also fabricate a solar cell based on a grating with a period of 615 nm, in which resonance modes were intentionally adjusted away from perovskite band edge (Supplementary Fig. 12). The EQE spectrum of this off-resonance (i.e. detuned) grating shows a comparable PV bandgap with its thin-film counterpart, thus highlighting the importance of optical resonances (Fig. 3e).

PV bandgap narrowing and $J_{sc}$ enhancement can be correlated to the GMRs in band edge regime by angle-resolved reflectance measurements of resonant solar cells (Supplementary Fig. 13). For resonant solar cells, suppression of reflection by GMRs can be achieved under both TE and TM polarizations. TE modes, which are lossy in perovskite gratings on the SiO$_2$ substrate, can be coupled into perovskites due to the existence of Ag mirror (Supplementary Fig. 13). Therefore, these resonant solar cells are insensitive to light polarization. Under TE polarization, BZ folding introduces an additional mode with a higher quality factor in supercell grating solar cells which is responsible for a narrower PV bandgap and a higher $J_{sc}$ compared to single-cell counterparts (Supplementary Fig. 13). Bandstructure calculations also reveal multiple GMRs in these resonant solar cells (Supplementary Figs. 14, 15), which is consistent with the angle-resolved spectra. Therefore, incorporation of multiple GMRs can boost strong absorption and efficient photocarrier generation to narrow PV bandgap and boost photocurrent. This resonant solar cell with a supercell grating has the potential to be applied to other perovskite compositions through the rational design of nanostructures. For instance, we can replicate the effects of giant band-edge light absorption, narrowed bandgap, and enhanced photocurrent using a MAPbI$_3$-based resonant solar cell (Supplementary Fig. 17).

**Photovoltaic performance of resonant solar cells**

We further compare the photovoltaic performance of solar cells based on these resonant structures and perovskite films. To evaluate the possibility of improving performance using a thicker perovskite, we fabricated perovskite films with thickness *t* of 900 and 1200 nm, which

are termed as thin and thick films respectively (Fig. 4a, b). In comparison, the resonant solar cell has a 930-nm-thick perovskite layer as presented in the cross-sectional SEM image (Fig. 4c). Thin-film solar cells present a maximum PCE of 23.0% with a $J_{sc}$ of 25.0 mA/cm$^2$ (Fig. 4d). With increasing *t* from 900 to 1200 nm, we observe the slightly improved $J_{sc}$ of 25.3 mA/cm$^2$ in thick-film solar cells, but both $V_{oc}$ and FF are compromised. This stems from poor crystallinity and limited carrier diffusion length in a thicker perovskite layer. Increasing perovskite thickness by 20–30% does not effectively boost $J_{sc}$ and yet it drops efficiency. On the other hand, resonant solar cells present a more feasible approach for improving PV performance. Resonant solar cells exhibit $V_{oc}$ of 1161 mV, $J_{sc}$ of 26.3 mA/cm$^2$ and FF of 80.0%, thus leads to a PCE of 24.4%. The $J_{sc}$ of 26.3 mA/cm$^2$ from *J-V* curve agrees well with EQE-integrated $J_{sc}$. The statistics on 16 devices for each type of solar cells confirm that the efficiency enhancement in resonant solar cells is attributed to increased $J_{sc}$ (Supplementary Fig. 18).

The shifting of band edge in resonant perovskite solar cells stems from the strong resonance modes. This effect can be clearly observed in light-emitting diodes as illustrated by the electroluminescence (EL) characterization of these three perovskite devices (Fig. 4e, f). We observe enhanced EL intensity and redshift of EL spectrum to 801 nm, arising from strong resonance modes in resonant solar cells. In striking contrast, the device based on perovskite films emits at 795 nm and weaker EL. The poorer EL in the thick film is correlated with its poorer crystallinity and higher defect density (Supplementary Fig. 19). This effect can be understood by the time-reversal symmetry between light absorption and emission[36], indicating that resonance modes for light absorption are also applicable to PL emission.

In conclusion, we demonstrate resonant perovskite solar cells via coupling multiple resonance modes to boost absorption at band edge. By rational design of a supercell grating, multiple GMRs are coupled into solar cells to achieve 18-nm redshift of band edge, thus leading to ~1.5 mA/cm$^2$ improvement of $J_{sc}$. Optical resonance provides a toolbox for sustaining multiple resonance modes to enhance light-matter

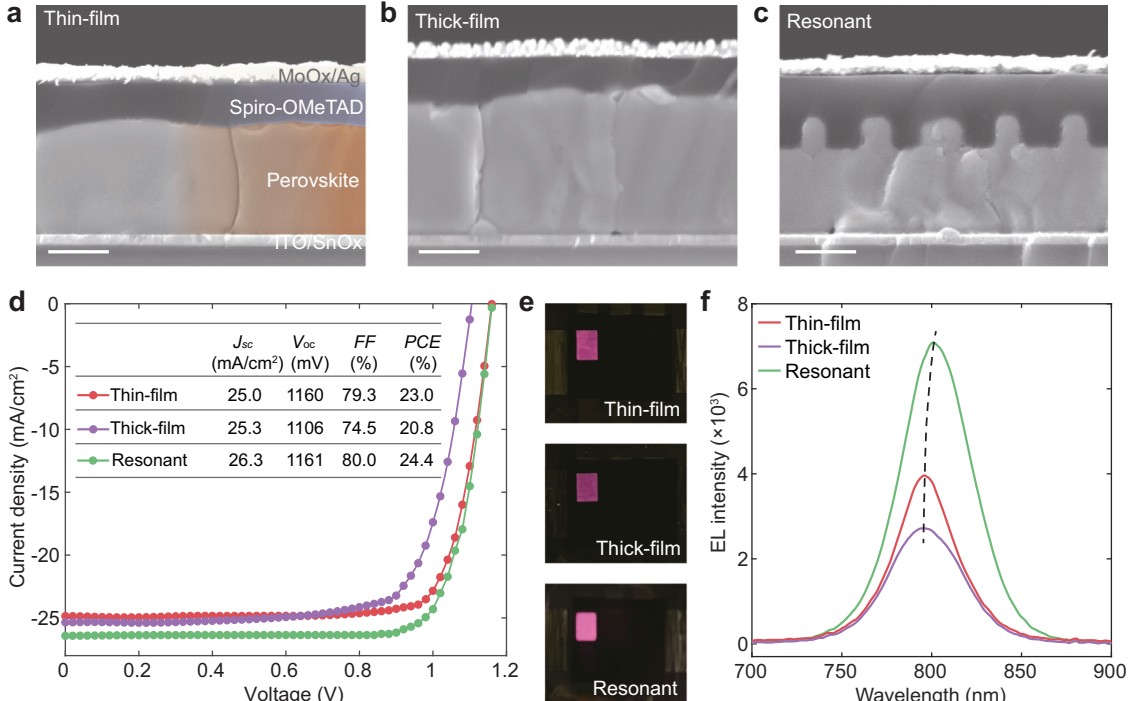

**Fig. 4 | Performances of resonant solar cells.** Cross-sectional SEM images of solar cells based on thin film with a thickness $t = 900$ nm (**a**), thick film with $t = 1200$ nm (**b**), and resonant nanostructures (**c**). Scale bars, 500 nm. **d** $J$-$V$ curves of solar cells. Inset shows short circuit current $J_{sc}$ (mA/cm²), open circuit voltage $V_{oc}$ (mV), fill factor $FF$ (%), and power conversion efficiency PCE (%) of solar cells. **e** Photographs of electroluminescence of thin-film (top), thick-film (middle), and resonant (bottom) solar cells. **f** EL spectra of solar cells. Dashed line is a guide to the eye.

interactions; our results highlight the power of coupling optical resonance with perovskites in extending the perovskite band edge. We believe that the findings in this work will provide a new route for increasing perovskite efficiencies after other strategies have reached their limits[37,38].

# Methods

## Materials

$SnO_2$ colloid precursor (tin (iv) oxide, 15% in $H_2O$ colloidal dispersion) was purchased from Alfa Aesar. Dimethylformamide (DMF), dimethyl sulfoxide (DMSO), Chlorobenzene (CB), isopropanol (IPA), 4-tert-butylpyridine (tBP), Spiro-OMeTAD, bis(trifluoromethane) sulfonimide lithium salt (Li-TFSI) and tris(2-(1H-pyrazol- 1-yl)-4-tert-butylpyridine)-cobalt(III)-tris(bis-(trifluoromethylsulfonyl) imide) (Co-FK209) were purchased from Sigma Aldrich. Lead iodine ($PbI_2$) was purchased from TCI. Formamidinium iodide (FAI), methylammonium chloride (MACl) and phenethylammonium iodide (PEAI) were purchased from Great Cell.

## Optical simulation

The finite-difference time-domain (FDTD) method is employed to simulate the optical absorption in perovskite films and resonant structures based on a commercially available software (Lumerical FDTD). The solar cells with glass, electrodes, electron/hole transport layers and perovskite layer are illuminated by a plane wave in -$z$ direction. Periodic boundary condition is applied in $x$ and $y$ direction to simulate the periodic grating structures. Perfectly matched layers (PMLs) are applied in $z$ direction to avoid reflection. The electric field $E$ in each layer is monitored and absorbed light power by perovskites can be expressed as $A = \int \omega \cdot \text{Im}(\varepsilon) \cdot |E|^2 \cdot dV$, where $\omega$ is the angular frequency of light, $\varepsilon$ is the permittivity of perovskite, $V$ is the volume of perovskite domain. The LHE is calculated by LHE = $A/P_{inc}$, where $P_{inc}$ is the incident light power. Photonic bandstructure is numerically simulated by eigenfrequency calculation in COMSOL Multiphysics. Unit cells of supercell and single-cell perovskite gratings are constructed with periodic boundary conditions in $x$ and $y$ directions and PMLs are applied in $z$ direction.

## Fabrication of resonant structures on perovskites

The perovskite resonant structures were fabricated by a thermal imprinting method with a negative replica silicon template. The silicon template was fabricated by standard nanofabrication processes. Firstly, ZEP-520A electron-beam resist was spin coated onto silicon and baked at 180 °C for 4 min. The designed patterns are exposed by using electron-beam lithography on Raith under 100 kV and 20 nA followed by developing in ZED-N50. 30 nm Cr was evaporated onto patterns of ZEP-520A resist in an electron-beam evaporation system (AJA International). The pattern was transferred onto Cr after lifting-off the resist and Cr served as a hard mask for inductively coupled plasma reactive-ion etching (ICP-RIE, Oxford Instrument) of silicon. After ICP-RIE, Cr hard mask was removed by hydrochloric acid. To prevent adhesion between perovskite and silicon template, a self-assembled monolayer of 1H,1H,2H,2H-perfluorooctyltriethoxysilane was decorated onto silicon surface by a vapor process at 90 °C. Thermal imprinting was performed in a EVG 501 wafer bonding system under $10^{-3}$ mbar vacuum. A perovskite thin film and a silicon template were combined and kept at a mechanical pressure of 200 bar and a temperature of 150 °C for 3 min.

## Device fabrication

Perovskite solar cells were fabricated with the following structure: indium tin oxide (ITO)/$SnO_2$/$FA_{0.95}Cs_{0.05}PbI_3$/Spiro-OMeTAD/$MoO_x$/Ag. ITO glass was cleaned by sequentially washing with detergent, deionized water, acetone and isopropyl alcohol. Firstly, ITO substrate was cleaned with ultraviolet ozone for 15 min. Subsequently, ITO substrate was spin coated with a layer of $SnO_2$ nanoparticle film (volume ratio of $SnO_2$ colloid precursor to water is 1:6) at 3000 r.p.m. for 30 s and annealed in ambient air at 150 °C for 30 min. Before transferring to glovebox, the substrate was treated with ultraviolet

ozone for 10 min. PbI₂/CsI solution was prepared by dissolving 1.4 mM PbI₂ and 0.07 mM CsI into 1 mL DMF/DMSO mixed solvent (v/v 94/6). FAI/MACl solution was prepared by dissolving 80 mg FAI and 13 mg MACl into 1 mL IPA. PbI₂ was spin coated on the substrate at 1500 rpm for 30 s. Subsequently, FAI/MACl solution was dynamically spin coated on the substrate at 1800 rpm for 40 s. It is worth noting that MACl is introduced to facilitate the crystallization of the perovskite phase but is easily volatized during the coating and annealing process[39,40]. Then the film was pre-annealed at 90 °C for 1 min inside glovebox and annealed outside of glovebox at 150 °C for 10 min under the humidity of ~35%. After annealing, thermal nanoimprinting was performed to transfer resonant structures onto perovskite thin films. A dynamic spin coating process was employed to apply a 2 mg/mL PEAI/IPA solution onto the perovskite layer at 4000 rpm for 30 s. It is noteworthy that dynamic coating potentially promotes a homogeneous distribution of the PEAI passivator in both the ridge and valley regions of the grating structure. Afterwards, spiro-OMeTAD solution [100 mg spiro-OMeTAD in 1.1 mL chlorobenzene with 39 µL tBP, 23 uL Li-TFSI (540 mg/mL in acetonitrile) and 10 µL Co-FK209 (376 mg/mL in acetonitrile)] was spin coated at 1750 rpm for 30 s. Finally, 5 nm MoOₓ and 100 nm Ag were thermally evaporated on the substrate sequentially under high vacuum to complete the whole device.

## Material characterizations

The morphology of perovskite resonant structures was characterized by SEM (Hitachi Regulus 8230). The refractive index $n$ and extinction coefficient $k$ were collected on a Semilab SE-2000 spectroscopic ellipsometer. Absorption spectra were recorded on Agilent Cary 7000 spectrophotometer. Absorption coefficients of perovskites were determined by measuring the reflectance and transmittance of perovskite thin films with calibrated thickness using SEM. XRD is recorded on an X-ray diffractometer (Malvern Panalytical) with Cu Kα radiation ($\lambda = 0.1542$ nm). TRPL measurements were conducted on a PicoQuant FluoTime 300 spectrometer. PL spectra were recorded on a Renishaw Raman and micro-PL system. Confocal PL mapping was measured on a Nikon A1 confocal microscope equipped with a 633 nm continuous-wave laser. Angle-resolved spectroscopy measurements were conducted on a home-built system with a Fourier imaging configuration. Reflectance signals were collected by a 50× objective with a numerical aperture of 0.75 and were detected by a Horiba iHR550 imaging spectroscopy with a 600 mm⁻¹ grating.

## PV measurements

The $J$–$V$ characteristics of the photovoltaic cells were obtained using a Keithley 2400 Source Meter under simulated one-sun AM 1.5 G illumination (100 mW/cm²) with a solar simulator (ABET TECHNOLOGIES, Sun 2000) in an ambient environment at 25 °C and ~50% relative humidity. The devices were measured in reverse scan with a 10 mV interval and 10 ms delay time. The EQE was measured using a Bentham EQE measurement system (PVE300-IVT). EL spectra of perovskite solar cells were measured with an integrated sphere and a spectrometer under an applied voltage of 1.2 V.

## Reporting summary

Further information on research design is available in the Nature Portfolio Reporting Summary linked to this article.

## Data availability

All data needed to evaluate the conclusions in the paper is present in the paper and/or the Supplementary Information.

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

## Acknowledgements

Y.H. acknowledges the support from MOE Tier 2 grant (MOE-T2EP10122-0005), the Ministry of Education (Singapore) and the National University of Singapore Presidential Young Professorship (A-0009174-03-00 and A-0009174-02-00). Authors of this paper are affiliated with the Solar Energy Research Institute of Singapore (SERIS), a research institute at the National University of Singapore. SERIS is supported by the National University of Singapore, the National Research Foundation Singapore, the Energy Market Authority of Singapore and the Singapore Economic Development Board. This research is supported by the National Research Foundation, Singapore, and A*STAR (Agency for Science, Technology and Research) under its LCERFI program Award No U2102d2002. R.S. gratefully acknowledges the funding support from the Singapore Ministry of Education via the AcRF Tier 2 grant (MOE-T2EP50222-0008) and Nanyang Technological University via Nanyang Assistant Professorship Start Up Grant.

## Author contributions

J.F. and Y.H. conceived the idea and designed the experiments. Y.H. directed and supervised the project. J.F. designed resonant structures and performed nanofabrication. J.F. and X.W. fabricated solar cells and conducted material and photovoltaic characterizations. W.W., J.F. and R.S. performed angle-resolved optical measurements. C.-W.Q. assisted with data analyses. J.L., H.L. and E.A. assisted with device fabrication and characterizations. J.F., X.W. and Y.H. analyzed the data and wrote the manuscript. All authors read and commented on the manuscript.

## Competing interests

The authors declare no competing interests.
