## [Peer Review File · Nature Communications]

Resonant Perovskite Solar Cells with Extended Band EdgeREVIEWER COMMENTS

Reviewer #1 (Remarks to the Author):

The work “Resonant Perovskite Solar Cells with Extended Band Edge” represents an interesting study of perovskite solar cell improvement based on a nanophotonic approach. The best solar cell demonstrates 24.4% efficiency, which is the result of J_{sc} growth because of absorption enhancement around the band edge. From the optical point of view, the work is solid and carried out on a high level. However, there are several critical drawbacks in the ‘device part’ making the work not acceptable in the current form:

1. There are only J-V parameters for the best cells, and the efficiency difference between the reference PSC and the improved one is only 1.4%. It might be just an error due to parameters fluctuation. The average values have to be included to the manuscript, as well as all statistical information should be provided with care. More than 10-20 devices should be checked to prove the existence of the grating-driven efficiency enhancement.
2. Lines 83-100: The authors discuss QS limit (better to give a full description) for PSCs with a simple formula. What about the record efficiency achieved for CsFAMAPb(IBr)₃ solar cells? Why is it ignored in the manuscript? I also suggest to add J-V parameters for each mentioned PSC.
3. FDTD calculations have been done for FACsPbI₃, the experimental part consists of the following perovskite materials: Formamidinium iodide (FAI), methylammonium chloride (MACl) and phenethylammonium iodide (PEAI). Why? What about the real perovskite composition? Please, include also XPS analysis.
4. PL peaks for all solar cells are not exactly at the same wavelength (fig 4 f), can the authors discuss more about the red shift? Actually, the thick film must have the highest PL signal if we assume that all materials have the same defects density. Also, according to SEM (fig 4 a-c), the cell with a grating structure has the thickest SPIRO layer. I also see that thick PSC has the lowest thickness of HTL. Also, there is an increased surface area between PVK and HTL. Maybe this is the answer to why you got the improved solar cell parameters.
5. Energy band structure should be calculated to provide more understanding on the barriers in the system.
6. Authors claim that they achieved to decrease band gap for FAPbI₃ and increase J_{sc} . However, they should also observe V_{oc} losses for the narrower perovskite band gap, but it does not happen. Why?
7. Regarding the literature review, there were plenty of other nanophotonic-based approaches used for the perovskite solar cells improvement (see e.g. [Advanced Photonics Research 3 (9), 2100326 (2022)] and many other works). Can the authors provide more comparisons with the other methods to prove the specific advantages of their grating-based method?

Reviewer #2 (Remarks to the Author):

The manuscript presents a novel wave-optics approach for boosting the efficiency of perovskite solar cells. Different from many other studies, their strategy doesn't create a structured Au electrode. They introduce a grating structure with folded Brillouin zone to generate guided-mode resonances near the perovskite band edge. This results in a band-edge extension, which contributes to an increased photocurrent with reduced thickness of perovskite. The method can potentially increase the efficiency of present-best perovskite solar cells. I recommend this manuscript for publication after minor revision.

- In Fig. 1 the authors show simulated light-harvesting efficiency (LHE) spectra, it's worthwhile giving the definition of LHE in the main manuscript.

- From the SEM in Fig. 4, the authors are using a thicker spiro layer in the resonant device structure compared to that in the non-resonant ones. Is the thickness modified for keeping the strongest resonance modes? Will the increased thickness limit the hole transport efficiency?

- A confocal PL image is presented in Fig. 2h, but it is not clear whether this measurement was performed on the supercell or single-cell grating. This should be clarified in the manuscript. Does the PL shift when different cell structures are employed?

- Is this method transferrable to other perovskite compositions to further increase the photocurrent without sacrificing Voc?

Reviewer #3 (Remarks to the Author):

Feng et al. incorporate resonant structures in FAPbI₃ based perovskite solar cell to enhance light absorption at band edge region. They designed effective resonant structure that maximizes the absorption at band edge region. With 18-nm band edge extension, 1.5 mA/cm² enhancement in J_{sc} was demonstrated. While light management strategies for perovskite solar cells have gained attention in recent years, this study reports systematic approach to design and fabricate microstructure that achieve a noticeable J_{sc} value of 26.0 mA/cm² with extended absorption region. I believe this study can provide useful insight to push the power conversion efficiency of perovskite solar cells toward thermodynamic limits. Thus, I recommend publication of this manuscript after addressing following comments.

- 1) In Fig. 2 h and supplementary Fig.8, it seems that the steady-state PL intensity has periodicity. Is this periodicity in PL intensity originated from structure? e.g. the peak region shows higher PL intensity while the valley region shows lower PL intensity. Is it related with stress applied during the nano imprinting process? Further explanation on detailed mechanism is required.
- 2) Related with comment 1, it seems that the nano imprinting process was performed after formation of perovskite layer. The mechanical stress applied to the perovskite layer should influence the quality of the perovskite layer (e.g. strain, cracking, grain growth etc). Further discussion on this aspect is necessary (e.g. <https://www.nature.com/articles/s41467-021-21803-2>).
- 3) For the EL data presented in Figure 4f, quantitative parameters should be presented (e.g. EL EQE). While the thin film and resonant structure demonstrate similar Vocs, the EL EQE of the resonant structure is much higher than that of the thin-film device. I guess this is related with the light out coupling efficiency.
- 4) In Figure 12b, it is interesting to see that the patterned perovskite film still maintain their grain boundaries. Can authors discuss the mechanism for structural change during the nanoimprinting process? Is it mediated through the residual solvents in the film? How thick was the film used for generating resonant structure in Figure 4c?
- 5) Is the anti-reflection coating was applied to the device? Near 95% EQE without anti-reflection coating is remarkable.
- 6) With patterned perovskite film, film top surface area inevitably increases. In this case, the surface passivation process might be critical. Can authors provide data and/or discussion on this point?

Response to reviewers' comments

Response to Reviewer 1:

Comment 1: The work “Resonant Perovskite Solar Cells with Extended Band Edge” represents an interesting study of perovskite solar cell improvement based on a nanophotonic approach. The best solar cell demonstrates 24.4% efficiency, which is the result of J_{sc} growth because of absorption enhancement around the band edge. From the optical point of view, the work is solid and carried out on a high level. However, there are several critical drawbacks in the ‘device part’ making the work not acceptable in the current form:

Response to Comment 1:

We sincerely appreciate Reviewer 1 for their positive evaluation of the optical aspects of our manuscript. We acknowledge the reviewer's feedback regarding the need to further strengthen the device-related part. In response to Reviewer 1's comments, we have made significant revisions to address these concerns. Firstly, we have included statistical analysis of solar cell performances, providing clear evidence that the enhanced PCE of resonant solar cells is attributed to an increase in J_{sc} . Secondly, we have used angle-resolved PL measurements to rationalize the redshift observed in the emission spectra of resonant solar cells is a result of improved light outcoupling. Thirdly, we have emphasized the correlation between bandgap and solar cell performance by incorporating J-V parameters and comparing our resonant solar cells with other nanophotonic structures, showcasing the advantages of our device operating in the wave-optics regime. We believe that these revisions significantly strengthen our manuscript and meets the requirements for publication in *Nature Communications*.

Comment 2: There are only J-V parameters for the best cells, and the efficiency difference between the reference PSC and the improved one is only 1.4%. It might be just an error due to parameters fluctuation. The average values have to be included to the manuscript, as well as all statistical information should be provided with care. More than 10-20 devices should be checked to prove the existence of the grating-driven efficiency enhancement.

Response to Comment 2:

We conducted measurements on 16 devices for each type of solar cells, and the statistics of parameters is presented in Fig. R1. The results reveal that resonant solar cells exhibit an improvement of 1-2 mA/cm² in J_{sc} compared to their thin-film counterparts, while maintaining comparable values of V_{oc} and FF . As a result, the average PCE shows a 1.4% increase in resonant solar cells. However, in the case of thick film, although there is a slight improvement in J_{sc} , the compromised V_{oc} and FF resulting from high defect density and poor crystallinity lead to a decrease in the average PCE to 20.2%.

We have included the statistics of devices in the revised Supplementary Information (Supplementary Fig. 18). Additionally, we have incorporated a relevant discussion in the revised manuscript as (Line 312-314),

The statistics on 16 devices for each type of solar cells confirm that the efficiency enhancement in resonant solar cells is attributed to increased J_{sc} (Supplementary Fig. 18).

Figure R1. Statistics of device performances. a-d, Statistics of J_{sc} (a), V_{oc} (b), FF (c), and PCE (d) based on 16 devices for each type of solar cells.

Comment 3: Lines 83-100: The authors discuss QS limit (better to give a full description) for PSCs with a simple formula. What about the record efficiency achieved for CsFAMAPb(IBr)₃ solar cells? Why is it ignored in the manuscript? I also suggest to add J-V parameters for each mentioned PSC.

Response to Comment 3:

Thank Reviewer 1's constructive suggestion on the inclusion of J - V parameters of mentioned solar cells. We have summarized state-of-the-art perovskite solar cells with different perovskite compositions (Table R1).

The correlation between perovskite composition and the S-Q limit efficiency is determined by material bandgap. The S-Q limit identifies an optimal bandgap of 1.34 eV, where a single-junction solar cell can achieve a maximum PCE of 33.7%. This optimal bandgap represents a balance between two competing factors: absorption and thermalization (*Science* 352, 307 (2016); *Nat. Rev. Mater.* 4, 269-285 (2019)).

Semiconductors with bandgaps below 1.34 eV experience thermalization, impeding the conversion of excess photon energy into electrical energy. Conversely, semiconductors with bandgaps above 1.34 eV suffer from lower thermodynamically limited efficiency due to the reduced absorption of photons below the bandgap energy.

Among lead-halide perovskites, FAPbI₃ demonstrates the narrowest bandgap of 1.48 eV, resulting in the highest photocurrents observed in perovskite solar cells. We acknowledge the viewpoint of Reviewer 1 that triple-cation CsFAMAPb(I/Br)₃ solar cells have previously achieved record efficiencies in the development of perovskite solar cells. However, with a better control of the phase stability, surface passivation, and crystal quality of FAPbI₃, recent advancements have shown that FAPbI₃-based perovskites achieve the highest J_{sc} and record PCEs (as shown in Ref. 1, 2, 6-9 in Table R1), compared to CsPbI₃ (Ref. 4), MAPbI₃ (Ref. 5) and triple-cation based solar cells. Therefore, our intention was not to overlook triple-cation perovskites, but rather to emphasize the tremendous potential for achieving high efficiency with perovskites featuring a narrower bandgap, as demonstrated by the recent developments in FAPbI₃-based solar cells.

To enhance our manuscript, we have incorporated Table R1 into the revised Supplementary Information (Supplementary Table 2). Furthermore, we have supplemented the revised manuscript with additional discussion as follows (Line 86-91).

The balance between photocarrier thermalization and sunlight absorption identifies an optimum bandgap of 1.34 eV in S-Q limit. For single-junction solar cells, GaAs with a bandgap of 1.42 eV produces a record efficiency of 29.1%¹. Likewise, among lead-halide perovskites, FAPbI₃ with the narrowest bandgap of 1.48 eV outperforms its perovskite counterparts, including MAPbI₃ and CsPbI₃ (Fig. 1a, Supplementary Table 2 for J - V parameters).

Table R1. State-of-the-art perovskite solar cells with different perovskite compositions. Subscript a and b denote PCE with and without certification, respectively.

Year	Perovskites	PCE (%)	V_{oc} (V)	FF (%)	J_{sc} (mA/cm ²)	Ref
2023	FAPbI ₃	25.73 _a	1.179	84.6	25.80	1
2022	FAPbI ₃	25.6 _a	1.182	82.7	26.30	2
2022	MAPbI ₃	22.52 _b	1.20	79.52	23.60	3
2022	CsPbI ₃	21.0 _b	1.20	84.1	20.86	4
2021	FA _{0.6} MA _{0.4} PbI ₃ single crystal	22.8 _b	1.1	0.79	26.2	5
2021	FAPbI ₃	25.5 _a	1.189	83.2	25.74	6
2021	FAPbI ₃	25.4 _a	1.177	81.5	26.28	7
2021	(FAPbI ₃) _{0.975} (MAPbBr ₃) _{0.025}	25.2 _a	1.181	84.8	25.14	8

2020	FAPbI ₃	24.64 _a	1.181	79.6	26.18	9
------	--------------------	--------------------	-------	------	-------	---

Reference

1. J. Park, J. Kim, H.-S. Yun, M. J. Paik, E. Noh, H. J. Mun, M. G. Kim, T. J. Shin, S. I. Seok, Controlled growth of perovskite layers with volatile alkylammonium chlorides. *Nature* **616**, 724-730 (2023).
2. Y. Zhao, F. Ma, Z. Qu, S. Yu, T. Shen, H.-X. Deng, X. Chu, X. Peng, Y. Yuan, X. Zhang, J. You, Inactive (PbI₂)₂RbCl stabilizes perovskite films for efficient solar cells. *Science* **377**, 531-534 (2022).
3. X. Zhuang, R. Sun, D. Zhou, S. Liu, Y. Wu, Z. Shi, Y. Zhang, B. Liu, C. Chen, D. Liu, H. Song, Synergistic Effects of Multifunctional Lanthanides Doped CsPbBrCl₂ Quantum Dots for Efficient and Stable MAPbI₃ Perovskite Solar Cells. *Adv. Funct. Mater.* **32**, 2110346 (2022).
4. S. Tan, B. Yu, Y. Cui, F. Meng, C. Huang, Y. Li, Z. Chen, H. Wu, J. Shi, Y. Luo, D. Li, Q. Meng, Temperature-Reliable Low-Dimensional Perovskites Passivated Black-Phase CsPbI₃ toward Stable and Efficient Photovoltaics. *Angew. Chem. Int. Ed.* **61**, e202201300 (2022).
5. A. Y. Alsalloum, B. Turedi, K. Almasabi, X. Zheng, R. Naphade, S. D. Stranks, O. F. Mohammed, O. M. J. E. Bakr, E. Science, 22.8%-Efficient single-crystal mixed-cation inverted perovskite solar cells with a near-optimal bandgap. *Energy Environ. Sci.* **14**, 2263-2268 (2021).
6. H. Min, D. Y. Lee, J. Kim, G. Kim, K. S. Lee, J. Kim, M. J. Paik, Y. K. Kim, K. S. Kim, M. G. Kim, T. J. Shin, S. Il Seok, Perovskite solar cells with atomically coherent interlayers on SnO₂ electrodes. *Nature* **598**, 444-450 (2021).
7. M. Kim, J. Jeong, H. Lu, T. K. Lee, F. T. Eickemeyer, Y. Liu, I. W. Choi, S. J. Choi, Y. Jo, H.-B. Kim, S.-I. Mo, Y.-K. Kim, H. Lee, N. G. An, S. Cho, W. R. Tress, S. M. Zakeeruddin, A. Hagfeldt, J. Y. Kim, M. Grätzel, D. S. Kim, Conformal quantum dot-SnO₂ layers as electron transporters for efficient perovskite solar cells. **375**, 302-306 (2022).
8. J. J. Yoo, G. Seo, M. R. Chua, T. G. Park, Y. Lu, F. Rotermund, Y.-K. Kim, C. S. Moon, N. J. Jeon, J.-P. Correa-Baena, V. Bulović, S. S. Shin, M. G. Bawendi, J. Seo, Efficient perovskite solar cells via improved carrier management. *Nature* **590**, 587-593 (2021).
9. M. Jeong, I. W. Choi, E. M. Go, Y. Cho, M. Kim, B. Lee, S. Jeong, Y. Jo, H. W. Choi, J. Lee, J.-H. Bae, S. K. Kwak, D. S. Kim, C. Yang, Stable perovskite solar cells with efficiency exceeding 24.8% and 0.3-V voltage loss. *Nature* **369**, 1615-1620 (2020).

Comment 4: FDTD calculations have been done for FACsPbI₃, the experimental part consists of the following perovskite materials: Formamidinium iodide (FAI), methylammonium chloride (MAcI) and phenethylammonium iodide (PEAI). Why? What about the real perovskite composition? Please, include also XPS analysis.

Response to Comment 4:

We acknowledge Reviewer 1's concern regarding the perovskite composition. Firstly, we would like to clarify that we conducted experimental measurements of the refractive index n and extinction coefficient k of the perovskite material (see Supplementary Fig. 1). **These measured values were then utilized in our FDTD calculations.** Hence, there is

no discrepancy in perovskite composition between the optical calculations and experimental data.

Figure R2. Evidence of no observable Cl⁻ in FAPbI₃ film. **a**, Cl 2p XPS spectra of FAPbI₃ fabricated with 10 mol% (MA-10), 30 mol% (MA-30), and 50 mol% (MA-50) MAI. Adapted from (*Joule* 3, 2179-2192 (2019)). **b**, ToF-SIMS depth profile of FAPbI₃ thin films fabricated from a precursor with 45 mol% MAI. Adapted from (*Nature* 616, 724-730 (2023)).

Secondly, we referred to the composition of the perovskites as FA_{0.95}CS_{0.05}PbI₃, considering the volatilization of MAI and the passivation function of PEAI. It is worth noting that MAI has been extensively studied for the crystallization of α -phase FAPbI₃, but it has been shown to be easily volatilized during the coating and annealing process (*Joule* 3, 2179-2192 (2019); *Nature* 616, 724-730 (2023)). The reported XPS and time-of-flight secondary ion mass spectrometry (ToF-SIMS) results demonstrate no observable Cl⁻ in the formed perovskite films (Fig. R2). As a consensus in the perovskite community, recent publications commonly refer to perovskite films fabricated using FAPbI₃ and MAI precursors as “FAPbI₃” (e.g. *Science* 369, 1615-1620 (2020); *Nature* 592, 381-385 (2021); *Science* 375, 302-306 (2022), etc.).

Regarding PEAI, we want to emphasize that we utilized it as a surface passivator. After the perovskite crystallization process, we dissolved PEAI in IPA and applied it as a coating onto the perovskite surface. PEAI, along with its phenyl-ammonium counterparts, has been widely investigated for perovskite surface passivation, which is only few nanometers thick at surface and does not impact the semiconductor bandstructure in bulk and simulation results. (*Nat. Photon.* 13, 460-466 (2019); *Nature* 598, 444-450 (2021); *Nat. Energy* 7, 229-237 (2022); *Nature* 603, 73-78 (2022)).

To avoid potential misunderstandings, we have included related discussion in revised manuscript (Line 386-387).

It is worth noting that MAI is introduced to facilitate the crystallization of the perovskite phase but is easily volatilized during the coating and annealing process^{39,40}.

Reference

39. Kim, M. et al. Methylammonium Chloride Induces Intermediate Phase Stabilization for Efficient Perovskite Solar Cells. *Joule* **3**, 2179-2192 (2019).

40. Park, J. et al. Controlled growth of perovskite layers with volatile alkylammonium chlorides. *Nature* **616**, 724-730 (2023).

Comment 5: PL peaks for all solar cells are not exactly at the same wavelength (fig 4f), can the authors discuss more about the red shift? Actually, the thick film must have the highest PL signal if we assume that all materials have the same defects density.

Response to Comment 5:

We appreciate Reviewer 1's comment regarding the redshift of EL peaks in the resonant solar cell. As also mentioned by Reviewer 3, redshift of PL/EL peaks in our manuscript can be understood as the enhanced *light outcoupling* through resonant nanostructures. The resonant modes can serve as “channels” for efficiently outcoupling of light from emitters to free space because **absorption and emission are two time-reversed processes**. In other words, resonant modes for boosting sunlight absorption also can enhance PL/EL emission.

To directly correlate the resonance modes to PL enhancement and redshift, we conducted angle-resolved PL measurements on the perovskite thin film, single-cell, and supercell gratings (Fig. R3). Compared to relative weak PL centered at 800 nm for the thin film, both single-cell and supercell gratings present bright PL from resonance modes at wavelengths above 800 nm, indicating that resonance modes provide channels for light outcoupling. These bright PL from resonance modes contribute to higher PL intensity at longer wavelengths, which is responsible to the spectral redshift and PL enhancement.

Figure R3. Angle-resolved PL spectra. a-c, Angle-resolved PL spectra of perovskite thin film (a), single-cell (b), and supercell (c) gratings. All samples were excited with a 457 nm continuous-wave laser at the same power.

Regarding the thick film, the weaker PL can be attributed to its higher defect density and poorer crystallinity compared to its thin-film and resonant-structure counterparts, as evidenced by the decreased V_{oc} and FF in the thick-film solar cell (Fig. 4d). To further support this observation, we conducted XRD and TRPL measurements (Fig. R4). In

comparison to its thin-film counterpart, the thick film exhibits a much stronger PbI_2 peak and poorer crystallinity of the perovskites, indicating incomplete reaction between PbI_2 and cations during the spin-coating and annealing process (Fig. R4a), which is line with previous reports (*Nature* 499, 7458 (2013)). The TRPL measurement of the thick film shows a fast component, which can be attributed to non-radiative recombination caused by defects. The PL lifetime, calculated as the time at $1/e$ of the maximum intensity, is decreased to 0.8 μs compared to 1.3 μs in the thin-film and resonant-structure counterparts. Our results on TRPL, XRD and solar cells are consistent with previous publications on thickness-dependent perovskite quality and device performances (such as *Org. Electron.* 21, 19-26 (2015); *J. Mater. Chem. C* 8, 36 (2020)).

Figure R4. XRD and TRPL of perovskite thick film. **a**, XRD diagrams of perovskite thin film and thick film. **b**, TRPL of perovskite thin film and thick film.

In summary, we attribute the enhancement and redshift of PL/EL in perovskite gratings to the presence of guided-mode resonances near the band edge. The poorer PL intensity observed in the thick film is correlated with its poor crystallinity and high defect density. To reinforce the findings presented in our manuscript, we have included these results in the revised Supplementary Information (Supplementary Fig. 10, 19). We also included discussion in the revised manuscript as follows.

Angle-resolved PL spectra further demonstrate bright PL emissions from the photonic bands at wavelengths above 800 nm (Supplementary Fig. 10). Therefore, we rationalize that this PL enhancement is a result of guided-mode resonances near the band edge, which sustain strong near fields and efficient photon emission channels³⁷. (Line 215-219)

The poorer EL in the thick film is correlated with its poor crystallinity and high defect density (Supplementary Fig. 18). (Line 320-322)

Comment 6: According to SEM (fig 4 a-c), the cell with a grating structure has the thickest SPIRO layer. I also see that thick PSC has the lowest thickness of HTL. Also, there is an increased surface area between PVK and HTL. Maybe this is the answer to why you got the improved solar cell parameters.

Response to Comment 6:

We appreciate this comment regarding the HTL thickness and increased interfacial area in resonant solar cells.

We would like to clarify that the spiro thickness is ranging from 290 nm to 380 nm in thin-film and thick-film solar cells. The difference in spiro thickness originates from the surface roughness of perovskite layers, which is difficult to be prevented for spin-coated perovskite films. In contrast, resonant solar cells demonstrate a more uniform spiro thickness around 330 nm at the ridge position of the grating, as depicted in Fig. R5. Therefore, based on the SEM images, we cannot conclude that there are variations in spiro thickness among different types of perovskite solar cells.

Figure R5. Cross-sectional SEM image of thin-film (a), thick-film (b) and resonant (c) solar cells. Scale bars, 500 nm. The arrows indicate the thickness at different positions.

We agree with Reviewer 1's point regarding the potential for improved device performance through increased contact area in resonant nanostructures, leading to enhanced carrier management. This enhancement is reflected in the FF and V_{oc} . However, we find that the resonant solar cell exhibits similar FF and V_{oc} compared to its thin-film counterpart, as evidenced by champion devices (Fig. 4d) and statistical analysis (Fig. R6). These results indicate that there is no significant alteration in carrier management, such as defect density, energy barrier, series resistance, and contact resistance, resulting from the larger surface area in resonant structures.

Figure R6. Statistics of V_{oc} (a), FF (b) based on 16 devices for each type of solar cells.

Comment 7: Energy band structure should be calculated to provide more understanding on the barriers in the system.

Response to Comment 7:

We appreciate Reviewer 1's suggestion to include the energy band structure in our study. To address this concern, we have provided the band alignment between the electrodes, electron/hole transport layers, and perovskites, as depicted in Fig. R5. The energy band structures of all materials, including electrodes, transport layers, perovskite layer (*Science*, 371, 636-640 (2021); *Sci. China Chem.* 65, 2299–2306 (2022)), were extensively studied in previous publications.

Figure R6. Band alignment of solar cell.

Comment 8: Authors claim that they achieved to decrease band gap for FAPbI₃ and increase J_{sc} . However, they should also observe V_{oc} losses for the narrower perovskite band gap, but it does not happen. Why?

Response to Comment 8:

We appreciate the comment on the narrower bandgap without V_{oc} losses. It is important to clarify that the perovskite bandgap is an intrinsic material property determined by its composition, phase, and strain. We did not claim to have achieved a narrower perovskite bandgap in our study. **The narrowed bandgap discussed in our manuscript refers to the photovoltaic (PV) bandgap, which is obtained from the EQE spectrum of a solar cell. This PV bandgap reflects the light absorption in the solar cell and is influenced by factors such as absorber layer thickness, interfacial reflection, and parasitic absorption.** In the Shockley-Queisser (S-Q) limit, the semiconductor band edge is simplified as a step function, assuming unity absorption for photon energies above the bandgap. However, real-world semiconductors often exhibit a Gaussian distribution of bandgap energies, leading to insufficient light absorption at the band edge (*Phys. Rev. Appl.* 7, 044016 (2017)). This effect is particularly pronounced in thin-film solar cells with submicrometer absorber layers (*Nat. Rev. Mater.* 4, 269-285 (2019)). For example, lead-halide perovskites have a narrowest bandgap of 1.48 eV, but

the PV bandgaps of record-efficiency solar cells range from 1.53-1.56 eV (*Science* 369, 1615-1620 (2020); *Science* 370, 108–112 (2020); *Nature* 590, 587-593 (2021); *Nature* 598, 444-450 (2021); *Science* 375, 302-306 (2022); *Nature* 616, 724-730 (2023)), indicating a shortfall in light absorption at the band edge.

Therefore, we sought to close this gap by designing strong optical resonances near band edge, which can enable a giant light absorption under a small absorption coefficient at perovskite band edge. This approach by enhancing band-edge photon absorption does not require the narrowing of material bandgap, thereby yielding a preserved V_{oc} and an enhanced J_{sc} .

Table R2. PV bandgap of solar cells and material bandgap of the absorber. Adapted from *Nat. Rev. Mater.* 4, 269-285 (2019).

Cell type (absorber)	E_g^{PV} (eV)	RT bandgap (eV)
c-Si	1.10	1.11(ref ¹)
GaAs	1.42	1.43 (ref ¹)
InP	1.38	1.35 (ref ²)
GaInP	1.88	NA
mc-Si	1.11	1.11(ref ¹)
CdTe _{1-x} Se _x	1.42	NA
CuIn _x Ga _{1-x} Se ₂	1.12	NA
Cu ₂ ZnSnS _{4-y} Se _y	1.18	NA
Cu ₂ ZnSnS ₄	1.48	NA
Perovskite	1.53-1.56	1.48
a-Si:H	1.77	1.69 (ref ³)
OPV (Toshiba)	1.62	NA
QD	1.77	NA

Comment 9: Regarding the literature review, there were plenty of other nanophotonic-based approaches used for the perovskite solar cells improvement (see e.g. [Advanced Photonics Research 3 (9), 2100326 (2022)] and many other works). Can the authors provide more comparisons with the other methods to prove the specific advantages of their grating-based method?

Response to Comment 9:

We appreciate Reviewer 1's comment regarding the existing nanophotonic approaches for solar cell improvement. To emphasize the unique advantages of our manuscript, we have conducted a careful examination of previous publications, and the results have been summarized in Table R3. Based on Table R3, we can highlight two main advantages of our manuscript.

First of all, our manuscript represents the first experimental demonstration of **resonant perovskite solar cells operated in wave-optics regime**. In striking contrast, previous publications on photonic design of perovskite solar cells belongs to **ray-optics approaches** by using textures for scattering¹⁻⁷ and diffraction⁸⁻¹⁰. To elongate optical path, textures underpin multiple scattering events and diffraction grating can deflect light to high angles. However, these strategies operate in ray-optics regime and thus provide limited enhancement, especially in spectral range with low absorption coefficient, because of the Yablonovitch limit of $4n^2$, where n is the refractive index of absorber (*J. Opt. Soc. Am.* 72, 899-907 (1982)). As a results, previous photonic design in perovskite solar cells cannot contribute photocurrent in the spectral region ($\lambda > 790$ nm) with small absorption coefficients, reflected by no extension of bandgap and limited J_{sc} of less than 24.5 mA/cm². Breaking ray-optics limit, we implement Brillouin-zone folding of guide-mode resonances to create resonant solar cells operating in wave-optics regime, which can solve the momentum mismatching between multiple waveguided modes and free-space light.

Secondly, our work firstly demonstrates **band-edge extension in submicrometer perovskite solar cells**. To the best of our knowledge, extension of PV band edge has been only observed in single-crystal solar cells with perovskite thickness larger than 10 μm (*Nat Commun* 8, 1890 (2017); *Energy Environ. Sci.* 14, 2263-2268 (2021)). However, such an approach cannot effectively improve the overall efficiency due to the compromised carrier transport in thick perovskite crystals, resulting in limited efficiencies below 23%. In our manuscript, we realize 18 nm redshift of perovskite band edge and a high J_{sc} of over 26.0 mA/cm² in submicrometer perovskite solar cells, which is comparable to state-of-the-art single-crystal solar cells. With greatly enhanced light absorption and preserved carrier management, we realize a high efficiency of 24.4%.

To strengthen our manuscript, we have added Table R3 with the references into revised Supplementary Information (Supplementary Table 1). We also added related discussion in Introduction part as (Line 78-81):

In comparison to ray-optics approaches employing texture and diffraction, the resonant perovskite solar cell manifests pronounced improvements in photocurrent and efficiency, attributed to its unprecedented giant band-edge light absorption (Supplementary Table 1).

Table R3 | Comparison of light-management strategies in solar cells. The subscript a represents EQE-integrated J_{sc} , and b represents J_{sc} extracted from IV scan.

Light-management strategy	Optics	J_{sc} (mA/cm ²)	Efficiency	Ref
Folded guided mode resonance	Wave optics	26.0 _a 26.3 _b	24.4%	Our work

Texture	Ray optics	22.0 _a 22.3 _b	16.3%	1
Texture	Ray optics	21.3 _b	17.7%	2
Texture	Ray optics	23.6 _b	19.8%	3
Texture	Ray optics	21.7 _b	17.1%	4
Texture	Ray optics	22.7 _a 23.5 _b	21.8%	5
Texture	Ray optics	23.1 _a 24.4 _b	22.2%	6
Texture	Ray optics	21.9 _b	16.3%	7
Diffraction	Ray optics	23.1 _b	19.7%	8
Diffraction	Ray optics	24.2 _b	19.6%	9
Diffraction	Ray optics	23.7 _b	21.8%	10

Reference

1. Pascoe, A. R. *et al.* Enhancing the optoelectronic performance of perovskite solar cells via a textured CH₃NH₃PbI₃ morphology. *Adv. Funct. Mater.* **26**, 1278-1285 (2016).
2. Dudem, B., Heo, J. H., Leem, J. W., Yu, J. S. & Im, S. H. CH₃NH₃PbI₃ planar perovskite solar cells with antireflection and self-cleaning function layers. *J. Mater. Chem. A* **4**, 7573-7579 (2016).
3. Zhang, H., Kramarenko, M., Osmond, J., Toudert, J. & Martorell, J. Natural Random Nanotexturing of the Au Interface for Light Backscattering Enhanced Performance in Perovskite Solar Cells. *ACS Photon.* **5**, 2243-2250 (2018).
4. Jošt, M. *et al.* Efficient Light Management by Textured Nanoimprinted Layers for Perovskite Solar Cells. *ACS Photon.* **4**, 1232-1239 (2017).
5. Wang, F. *et al.* Coordinating light management and advance metal nitride interlayer enables MAPbI₃ solar cells with >21.8% efficiency. *Nano Energy* **92**, 106765 (2022).
6. Tavakoli, M. M. *et al.* Ambient stable and efficient monolithic tandem perovskite/PbS quantum dots solar cells via surface passivation and light management strategies. *Adv. Funct. Mater.* **31**, 2010623 (2021).
7. Wei, J. *et al.* Enhanced Light Harvesting in Perovskite Solar Cells by a Bioinspired Nanostructured Back Electrode. *Adv. Energy Mater.* **7**, 1700492 (2017).
8. Wang, Y. *et al.* Diffraction-Grated Perovskite Induced Highly Efficient Solar Cells through Nanophotonic Light Trapping. *Adv. Energy Mater.* **8**, 1702960 (2018).
9. Deng, K., Liu, Z., Wang, M. & Li, L. Nanoimprinted Grating-Embedded Perovskite Solar Cells with Improved Light Management. *Adv. Funct. Mater.* **29**, 1900830 (2019).
10. Wang, Y. *et al.* Colorful efficient moiré-perovskite solar cells. *Adv. Mater.* **33**, 2008091 (2021).

Response to Reviewer 2:

Comment 1: The manuscript presents a novel wave-optics approach for boosting the efficiency of perovskite solar cells. Different from many other studies, their strategy doesn't create a structured Au electrode. They introduce a grating structure with folded Brillouin zone to generate guided-mode resonances near the perovskite band edge. This results in a band-edge extension, which contributes to an increased photocurrent with reduced thickness of perovskite. The method can potentially increase the efficiency of present-best perovskite solar cells. I recommend this manuscript for publication after minor revision.

Response to Comment 1:

We sincerely appreciate Reviewer 2 for their positive evaluation and recommendation of publication with minor revisions. In response to their constructive comments and suggestions, we have performed angle-resolved PL experiments to further investigate the observed PL redshift. Additionally, we have conducted FDTD simulations to demonstrate the transferability of the resonant solar cell concept to other perovskite compositions. We are confident that the revised version of our manuscript adequately addresses these concerns and meets the high standards required for publication in *Nature Communications*.

Comment 2: In Fig. 1 the authors show simulated light-harvesting efficiency (LHE) spectra, it's worthwhile giving the definition of LHE in the main manuscript.

Response to Comment 2:

We appreciate Reviewer 2 for this constructive suggestion. In Method section, we have discussed how to numerically calculate LHE via FDTD method but did not include the physical definition of LHE.

LHE represents the percentage of incident photons that are effectively absorbed by the absorber layer of the solar cell. In our study, LHE is specifically defined as the fraction of photons absorbed by the perovskite layer, while excluding photons that are reflected or absorbed by other layers such as electrodes, HTL, ETL, and glass substrate. The relationship between LHE and EQE can be written as (*Acc. Chem. Res.* 42, 1788-1798 (2009)),

$$EQE(\lambda) = LHE(\lambda)\phi_{inj}\eta_e \quad (1)$$

where ϕ_{inj} is the charge-injection quantum efficiency and η_e is the charge collection efficiency. Hence, LHE is equal to EQE under the assumption of ideal absorbed photon to charge carrier conversion, as well as ideal charge injection and collection. To strengthen our manuscript, we have included following discussion in our revised Manuscript (Line 108-113).

In our simulations, LHE is defined as the fraction of photons absorbed by the perovskite layer, while excluding photons that are reflected or absorbed by other layers such as electrodes, transport layers, and glass. It is important to note that LHE is conceptually equivalent to EQE, under the assumption of ideal conversion of absorbed photons to charge carriers, as well as ideal charge injection and collection³³.

Reference

33. Grätzel, M. Recent Advances in Sensitized Mesoscopic Solar Cells. *Acc. Chem. Res.* **42**, 1788-1798 (2009).

Comment 3: From the SEM in Fig. 4, the authors are using a thicker spiro layer in the resonant device structure compared to that in the non-resonant ones. Is the thickness modified for keeping the strongest resonance modes? Will the increased thickness limit the hole transport efficiency?

Response to Comment 3:

We appreciate this comment regarding the spiro-OMeTAD thickness. As also mentioned by Reviewer 1, we conducted a thorough comparison of spiro thicknesses across different types of solar cells. Our analysis revealed that thin-film and thick-film solar cells exhibit a wide range of spiro thicknesses (290-370 nm), whereas the resonant solar cell demonstrates a more uniform spiro capping layer thickness of about 330 nm. Therefore, we cannot conclude that the resonant solar cell has a thicker spiro layer. Regarding the correlation between spiro thickness and optical resonances, it is important to clarify that while spiro thickness can influence the resonance wavelength, its effect is relatively smaller compared to perovskites. This is primarily due to the lower refractive index of spiro (1.62) in contrast to perovskites (2.5).

Figure R7. A published *n-i-p* perovskite solar cell with a 350-450 nm thick spiro-OMeTAD layer, which achieved a certified efficiency of 25.6%. The SEM image is reproduced from (*Science* **377**, 531-534 (2022)).

The efficiency of hole transport in solar cells can be evaluated by the fill factor (*FF*). Our *J-V* results (Figure 4d, Supplementary Figure 18) demonstrate that the resonant solar cell

exhibits a comparable FF to its thin-film counterpart, indicating efficient hole transport. This finding suggests that the implementation of a 330 nm thick spiro layer in the resonant solar cell does not compromise the hole transport efficiency. To further emphasize the potential of a thick spiro layer for efficient hole transport, we present another SEM image of a previously published $n-i-p$ device with a certified efficiency of 25.6%, which featured a spiro layer thickness ranging from 350 to 450 nm (*Science* 377, 531-534 (2022)). This additional evidence supports the efficient hole transport capability of thick spiro layers.

Comment 4: A confocal PL image is presented in Fig. 2h, but it is not clear whether this measurement was performed on the supercell or single-cell grating. This should be clarified in the manuscript. Does the PL shift when different cell structures are employed?

Response to Comment 4:

We appreciate this comment. The confocal PL images, PL spectra and XRD that were labelled as “resonant” are performed on the supercell grating. To avoid potential misunderstandings, we added the discussion in the caption of Fig. 2 as,

“Resonant” nanostructure refers to the supercell grating.

We appreciate your valuable feedback regarding the PL redshift. To provide a direct correlation between the PL redshift and resonance modes, as well as to differentiate the PL emissions between single-cell and supercell gratings, we performed angle-resolved PL measurements, as shown in Figure R8. Firstly, for both single-cell and supercell gratings, we observe bright PL emissions from photonic bands at wavelengths longer than 800 nm, which are responsible for the redshift of the PL spectra compared to their thin-film counterpart. Secondly, we find that the supercell grating exhibits a higher photonic density of states compared to its single-cell counterpart. This is due to the Brillouin-zone (BZ) folding, which introduces more photonic bands from high momenta to the BZ center. As a result, the supercell grating exhibits a higher PL intensity at wavelengths above 800 nm. To further strengthen our manuscript, we have included results of the angle-resolved PL measurements in the revised Supplementary Information (Supplementary Fig. 10).

Figure R8. Angle-resolved PL spectra. a-c, Angle-resolved PL spectra of perovskite thin film (a), single-cell (b), and supercell (c) gratings. All samples were excited with a 457 nm continuous-wave laser at the same power.

Comment 5: Is this method transferrable to other perovskite compositions to further increase the photocurrent without sacrificing Voc?

Response to Comment 5:

We appreciate Reviewer 2's comment on the transferability of our method. Unlike carrier-management methods that are sensitive to the chemical composition of perovskites, our resonant solar cells can be directly transferred to other perovskite materials by rationally designing resonant nanostructures.

To demonstrate our advantage, we conducted optical simulations based on MAPbI₃ solar cells. Firstly, we simulated the LHE spectrum of a thin-film solar cell with an 800 nm thick perovskite layer (Fig. R9e, f). The PV bandgap is determined as 1.592 eV (779 nm), consistent with the reported EQE measurement results of MAPbI₃ solar cells (*Nat. Commun.* 11, 5402 (2020)). Secondly, we designed a resonant MAPbI₃ solar cell with a supercell grating (Figure R9d). The simulated dispersion of LHE spectra as a function of grating period reveal multiple guided-mode resonances near the perovskite band edge under both TM and TE polarizations (Figure R9a-c). At a period of 528 nm, we observe a PV bandgap of 1.553 eV, corresponding to 39 meV narrower bandgap and 19.5 nm redshift of LHE spectrum compared to its thin-film counterpart (Figure R9e, f). These results highlight the potential to reproduce giant band-edge light absorption, band-edge extension and enhanced photocurrent with other perovskite compositions.

To strengthen our manuscript, we have included the simulation results in revised Supplementary Information (Supplementary Fig. 17) and related discussion has been added in revised main text as (Line 284-288),

This resonant solar cell with a supercell grating has the potential to be applied to other perovskite compositions through the rational design of nanostructures. For instance, we can replicate the effects of giant band-edge light absorption, narrowed bandgap, and enhanced photocurrent using a MAPbI₃-based resonant solar cell (Supplementary Fig. 17).

Figure R9. Simulations of a resonant perovskite solar cell based on MAPbI₃. **a-c**, FDTD simulations of LHE spectra as a function of grating period under TM (**a**), TE (**b**), and unpolarized (**c**) light. **d**, Device configuration for the optical simulation of the resonant perovskite solar cell utilizing a supercell grating. The parameters are $t_s = 280$ nm, $t_g = 250$ nm, and $t_w = 660$ nm. The refractive index of MAPbI₃ is extracted from *Opt. Express* 26, 27441 (2018). **e**, Simulated LHE spectra of thin-film and resonant perovskite solar cells. The period of the supercell grating is 528 nm. **f**, Simulated $dLHE/d\lambda$ spectra of thin-film and resonant perovskite solar cells, illustrating the extension of the band edge after introducing optical resonances.

Response to Reviewer 3:

Comment 1: Feng et al. incorporate resonant structures in FAPbI₃ based perovskite solar cell to enhance light absorption at band edge region. They designed effective resonant structure that maximizes the absorption at band edge region. With 18-nm band edge extension, 1.5 mA/cm² enhancement in J_{sc} was demonstrated. While light management strategies for perovskite solar cells have gained attention in recent years, this study reports systematic approach to design and fabricate microstructure that achieve a noticeable J_{sc} value of 26.0 mA/cm² with extended absorption region. I believe this study can provide useful insight to push the power conversion efficiency of perovskite solar cells toward thermodynamic limits. Thus, I recommend publication of this manuscript after addressing following comments.

Response to Comment 1:

We appreciate the constructive comments and high evaluations provided by Reviewer 3. In response to Reviewer 3's suggestions, we have conducted angle-resolved PL measurements to establish a correlation between the redshift of PL/EL spectra and light outcoupling facilitated by guided-mode resonances. We believe that the revisions made to the manuscript have adequately addressed the concerns raised and meet the standards for publication in *Nature Communications*.

Comment 2: In Fig. 2h and supplementary Fig.8, it seems that the steady-state PL intensity has periodicity. Is this periodicity in PL intensity originated from structure? e.g. the peak region shows higher PL intensity while the valley region shows lower PL intensity. Is it related with stress applied during the nano imprinting process? Further explanation on detailed mechanism is required.

Response to Comment 2:

We appreciate this comment regarding the confocal PL imaging. We agree with Reviewer 1 that the observed periodicity in PL intensity is indeed due to the structural characteristics of the gratings. However, it should not be attributed to reduced PL quantum efficiency in valley region.

The brighter PL emission from grating ridges can be attributed to the principle of confocal PL imaging, in which a pinhole is applied to eliminate out-of-focus light signals. Therefore, only light very close to focal plane can be detected and this selective detection enables high spatial resolution of confocal PL. In our study, the perovskite gratings have a height difference of 250 nm between the ridge and valley regions. We chose the grating ridge as the focal plane to enable the spatial resolution of PL imaging, thus leading to a brighter PL signal from the ridges compared to the valleys.

To illustrate the impact of focal plane to confocal imaging and exclude the PL inhomogeneity in nanoimprinted grating, we conducted confocal PL imaging of a

supercell grating with the focal plane set in the valley region. In this region, we observe a relatively homogeneous PL intensity, attributed to eliminated height difference and similar PL quantum efficiency (Figure R10).

In conclusion, the observed periodicity in the PL imaging is attributed to the principle of confocal PL rather than the inhomogeneous PL resulting from applied stress during the nanoimprint process. To avoid potential misunderstanding, we have included related discussion in revised Supplementary Information (Supplementary Fig. 8) as follows.

The periodicity in the confocal PL imaging is attributed to the principle of confocal PL imaging, in which a pinhole is applied to eliminate out-of-focus light signals. The grating ridge was chosen as the focal plane to enable the spatial resolution of PL imaging, thus leading to a brighter PL signal from the ridges compared to the valleys.

Figure R10. Confocal PL imaging of a supercell grating with the focal plane at grating ridge (a), and valley (b). Scale bars, 5 μm .

Comment 3: Related with comment 1, it seems that the nano imprinting process was performed after formation of perovskite layer. The mechanical stress applied to the perovskite layer should influence the quality of the perovskite layer (e.g. strain, cracking, grain growth etc). Further discussion on this aspect is necessary (e.g. <https://www.nature.com/articles/s41467-021-21803-2>).

Response to Comment 3:

We appreciate the comment regarding applied mechanical stress and would like to thank Reviewer 3 for providing the insightful reference. To evaluate the strain in perovskites before and after nanoimprinting, we conducted X-ray diffraction (XRD) analysis. As shown in Fig. R11, no observable shift or change in linewidth was observed after the nanoimprinting process for the formation of single-cell and supercell gratings. This indicates that there was no net lattice strain and no significant change in the crystallinity of the perovskite resonant nanostructures. These findings are consistent with the TRPL results, which demonstrate comparable PL lifetimes for the perovskite thin film and the resonant nanostructures (Fig. 2i). We attribute the absence of observable lattice strain to the post-crystallization nanoimprinting process. Nanoimprinting was

performed after annealing the perovskites at 150 °C for 10 minutes, ensuring complete crystallization of the material.

Secondly, we acknowledge the possibility of a local strain distribution within the nanoimprinted resonant structures, similar to the reported variations in photoluminescence lifetimes and photocurrents between the hills and valleys of wrinkled perovskite films (*Nat. Commun.* **12**, 1554 (2021)). However, due to the diffraction limit of the optical system, it is impossible to resolve ridge and valley regions of our sub-wavelength nanostructures in fluorescence lifetime imaging microscopy. Furthermore, it is extremely challenging for performing photoconductive AFM or kelvin probe force microscopy (KPFM) on our resonant nanostructures with a large aspect ratio (200-300 nm in width and 250 nm in height). Therefore, we regret that we cannot provide further evidence of the strain distribution in the resonant nanostructure.

To strength our manuscript, we have made the following revisions (Line 211-213):

The material quality improvement or lattice strain³⁸ for stronger PL can be ruled out by time-resolved PL measurements, which shows similar PL lifetimes for the perovskite thin film and resonant structure (Fig. 2i).

Reference

38. Kim, S.-G. et al. How antisolvent miscibility affects perovskite film wrinkling and photovoltaic properties. *Nat. Commun.* **12**, 1554 (2021).

Figure R11. XRD diagram of (100) plane of perovskite thin film, single-cell and supercell gratings.

Comment 4: For the EL data presented in Figure 4f, quantitative parameters should be presented (e.g. EL EQE). While the thin film and resonant structure demonstrate similar

Vocs, the EL EQE of the resonant structure is much higher than that of the thin-film device. I guess this is related with the light out coupling efficiency.

Response to Comment 4:

We agree that both EL and PL enhancement in the resonant solar cell is a result of improved light outcoupling efficiency. The observed PL enhancement in confocal PL imaging (Fig. 2g, h, Supplementary Fig. 8) and PL spectra (Supplementary Fig. 9) can also be attributed to enhanced light outcoupling in resonant nanostructure.

Due to the unavailability of EL EQE measurements at the moment, instead we have performed angle-resolved photoluminescence spectroscopy to analyze the angle-dependent photonic band structure of the emission. In Figure R10a, weak PL with weak angle dependence is observed from a thin film. However, in striking contrast, bright PL emissions from photonic bands are observed from both single-cell and supercell gratings at wavelengths above 800 nm (Figure R10b, c). This observation indicates that these photonic bands serve as "channels" for outcoupling PL from perovskite emitters to free space. The presence of these photonic bands, characterized by PL, is also observed in angle-resolved reflectance measurements of resonant gratings (Fig. 2e, f) and resonant solar cells (Supplementary Fig. 14). The similarity between the resonant photonic bands observed in reflection and emission are mainly attributed to the fact that **absorption and emission are two time-reversed processes**. In other words, the resonant nanostructures designed for absorption enhancement can also facilitate PL and EL emission.

To strengthen our manuscript, we have included the angle-resolved PL in revised Supplementary Information and added following discussion in main text (Line 215-219).

Angle-resolved PL spectra further demonstrate bright PL emissions from the photonic bands at wavelengths above 800 nm (Supplementary Fig. 10). Therefore, we rationalize that this PL enhancement is a result of guided-mode resonances near the band edge, which sustain strong near fields and efficient photon emission channels³⁸.

Figure R12. Angle-resolved PL spectra. a-c, Angle-resolved PL spectra of perovskite thin film (a), single-cell (b), and supercell (c) gratings. All samples were excited with a 457 nm continuous-wave laser at the same power.

Comment 5: In Figure 12b, it is interesting to see that the patterned perovskite film still maintain their grain boundaries. Can authors discuss the mechanism for structural change during the nanoimprinting process? Is it mediated through the residual solvents in the film? How thick was the film used for generating resonant structure in Figure 4c?

Response to Comment 5:

We appreciate this comment on the structure formation mechanism during nanoimprinting. However, it's important to note that we did not employ a thin film with residual solvents for nanoimprinting. Prior to the nanoimprinting process, we annealed the perovskite film outside of a glovebox at 150°C for 10 minutes to ensure the complete removal of solvents and the crystallization of perovskites. This annealing process is consistent with the perovskite fabrication recipe for thin-film devices.

Figure R13. SEM image of the perovskite thin film used for the fabrication of resonant nanostructures and solar cells. Scale bar, 1 μm .

To assess the quality and thickness of the perovskite layer prior to nanoimprinting, we performed cross-sectional SEM. In Figure R13, we observe perovskite grains with a large size and a thickness of about 800 nm. Considering the average duty ratio of 0.44 for the supercell grating and a total perovskite thickness of 930 nm in the resonant solar cell, the estimated perovskite film thickness before nanoimprinting is $930 - 250 * 0.44 = 820$ nm, which aligns well with our SEM results.

Therefore, we attribute the formation of perovskite resonant structures to the applied mechanical pressure of 200 bar. This pressure induces the deformation of the thin film and enables the conformal transfer of the designed patterns from the template to the perovskite layer. Similar methods have been successfully employed in the fabrication of perovskite nanostructures for enabling lasing emission (*Adv. Mater.* 29, 1605003 (2017); *Adv. Mater.* 35, 2207430 (2023)), demonstrating the capability of this approach for precise and efficient fabrication of resonant nanostructures.

Comment 6: Is the anti-reflection coating was applied to the device? Near 95% EQE without anti-reflection coating is remarkable.

Response to Comment 6:

A commercial ARC coating was applied to reduce ~2% reflection at air-glass interface.

Comment 7: With patterned perovskite film, film top surface area inevitably increases. In this case, the surface passivation process might be critical. Can authors provide data and/or discussion on this point?

Response to Comment 7:

In our manuscript, we implement PEAI to passivate the surface of nanoimprinted perovskites. To enable a homogeneous distribution of PEAI on a nanostructured perovskite surface, we conducted a dynamic spin coating of PEAI solution at 4000 rpm for 30s. It worths to note that this dynamic coating can prevent the inhomogeneous distribution of passivators in the ridges and valleys of grating.

To strengthen our manuscript, we have included the following discussion in the Method section of revised manuscript (Line 391-394),

A dynamic spin coating process was employed to apply a 2 mg/mL PEAI/IPA solution onto the perovskite layer at 4000 rpm for 30 s. It is noteworthy that dynamic coating potentially promotes a homogeneous distribution of the PEAI passivator in both the ridge and valley regions of the grating structure.

REVIEWERS' COMMENTS

Reviewer #1 (Remarks to the Author):

The authors have addressed all my comments, and the manuscript is ready for acceptance in my opinion.

Reviewer #2 (Remarks to the Author):

The authors address each of my comments and add sufficient evidence to strengthen the results and conclusion. I recommend the publication of the revised manuscript.

Reviewer #3 (Remarks to the Author):

Authors well addressed all the comments by the reviewers. Therefore, I recommend publication of this manuscript.